# Isotope-derived young water fractions in streamflow across the tropical Andes mountains and Amazon floodplain

Emily I. Burt[1,*], Daxs Herson Coayla Rimachi [2,3], Adan Julian Ccahuana Quispe[2], Abra Atwood[1], A. Joshua West[1]

[1]Department of Earth Sciences, University of Southern California, Los Angeles, California, USA 90089.
[2]Universidad Nacional San Antonio Abad del Cusco (UNSAAC), Cusco, Peru.
[3]Universidad Científica del Sur, Lima, Peru.
*Now at Chapman University, Schmid College of Science and Technology, Orange, California, USA 92866.

*Correspondence to:* Emily I. Burt, emburt@chapman.edu

## Abstract

The role of topography in determining water transit times and pathways through catchments is unclear, especially in mountainous environments — yet these environments play central roles in global water, sediment, and biogeochemical fluxes. Since the vast majority of intensively monitored catchments are in northern latitudes, the interplay between water transit, topography and other landscape and climatic characteristics is particularly underexplored in tropical environments. To address this gap, here we present the results of a multi-year hydrologic sampling campaign (twice monthly and storm sampling) to quantify water transit in seven small catchments ($< 1.3$ km$^2$ area) across the transition from the Andes mountains to the Amazon floodplain in southern Peru. We use the stable isotope composition of water ($\delta^{18}O$) to calculate the fraction of streamflow comprised of recent precipitation ("young water fraction") for each of the seven small catchments. Flow-weighted young water fractions ($F_{yw}$) are 5-26 % in the high-elevation mountains, 22-52 % for mid-elevation mountains and 7 % in the foreland floodplain. Across these catchments, topography does not exert a clear control on water transit; instead stream $F_{yw}$ is apparently controlled by a combination of hydroclimate (precipitation regime) and bedrock permeability. Mid-elevation sites are posited to have the highest $F_{yw}$ due to more frequent and intense rainfall; less permeable bedrock and poorly developed soils may also facilitate high $F_{yw}$ at these sites. Lowland soils have low $F_{yw}$ due to very low flow path gradients despite low permeability. The data presented here highlight the complexity of factors that determine water transit in tropical mountainous catchments, particularly highlighting the role of intense orographic precipitation at mountain fronts in driving rapid conveyance of water through catchments. These results have implications for the response of Earth's montane "water towers" to climate change and for water-rock reactions that control global biogeochemical cycles.

## 1.    Introduction

The time that water takes to transit through a watershed provides a measure of the age of water as it leaves the system (*sensu* Benettin et al. 2022) and has important physical, chemical, and biological implications. Water transit times through catchments can influence how freshwater resources respond to a changing climate. Catchment transit times can also inform understanding of how water partitions into different fluxes (e.g., discharge vs. evapotranspiration), with implications for determining the age (or seasonal origin) of water used by vegetation (Allen et al., 2019; Kirchner and Allen, 2020; Rempe and Dietrich, 2018). And, transit times are thought to influence water quality and regulate rates of chemical weathering, and thus the alkalinity fluxes that control the geologic carbon cycle, because the flow of water can modulate the saturation state of fluids with respect to mineral weathering reactions (Ameli et al., 2017; Berner, 1978; Maher, 2010, 2011). Mountain watersheds often serve as sources of water, solute and sediment fluxes to lowland ecosystems (Barnett et al., 2005; Gaillardet et al., 1999; Immerzeel et al., 2020; Viviroli et al., 2007), and mountain environments are especially sensitive to changing climate – emphasizing the particular importance of understanding water transit times in these settings. Tropical montane watersheds, in particular, play outsized roles in global water and biogeochemical cycles (Fekete et al., 2002; Meybeck, 1987), yet are underrepresented in studies of terrestrial water transit.

Previous research has suggested that topography, subsurface structure, and hydroclimate all play roles in determining the ages of water moving through mountainous watersheds. Some studies indicate that steeper slopes and shorter flowpaths generally lead to faster water transit (McGlynn et al., 2003; McGuire et al., 2005; Tetzlaff et al., 2009), while others have found that steeper watersheds have slower water transit than more gently sloping catchments (Jasechko, 2016; Lutz et al., 2018). Other work has shown how the permeability of soil and bedrock, and the associated depth of active flow paths, can influence water transit times (Hale et al., 2016; Hale and McDonnell, 2016; Xiao et al., 2021; Asano and Uchida, 2012; Muñoz-Villers et al., 2016a). Finally, the timing and amount of precipitation can influence water transit through mountain landscapes (Gallart et al., 2020a; Stockinger et al., 2019; von Freyberg et al., 2018a; Wilusz et al., 2017). Disentangling the effects of topography, subsurface structure, and hydroclimate remains challenging, and few studies have tested ideas about the controls on water transit across the dramatic topographic gradients of major mountain ranges, i.e., from steep high relief mountain catchments to the surrounding floodplains.

The stable isotopic composition of stream and river water is a conservative tracer that provides powerful information about catchment transit times. For decades, many studies used time series of stable isotopes of

water in streams and precipitation to calculate stream mean transit times or stream water "ages"

(Małoszewski and Zuber, 1982; McGuire and McDonnell, 2006), but these calculations were prone to errors arising from aggregation biases and hydrologic non-stationarity (Kirchner, 2016a, b). Instead, recent work focuses on using stable isotope time series to calculate the fraction of water within a reservoir comprised of recent precipitation (e.g., precipitation that fell within the 2-3 preceding months). The fraction of recent precipitation is referred to as the young water fraction ($F_{yw}$) and is calculated from the ratio

between the amplitude of the seasonal cycle of oxygen isotopes in water from streamflow and precipitation (Kirchner, 2016a, b). The young water fraction offers information about water transit over timescales relevant to many hydrologic problems, making it an appropriate tool for untangling the impacts of topography, subsurface structure, and hydroclimate on water transit in tropical watersheds.

The primary objective of this work was to test the factors controlling young water fractions in streamflow across a dramatic mountain-to-floodplain gradient in the tropics. We collected a five-year time series (2016−2020) of approximately fortnightly stream and precipitation water isotope samples from seven small (< 1.3 km²) catchments in southern Peru spanning an elevation range of more than 3200 m and a gradient in catchment slopes from 38 ° to 4 ° in order to ask the following questions: 1) how do precipitation and

stream water isotopes vary seasonally, and how do they differ from one another, among different catchments in this setting?, and 2) how does the young water fraction vary among catchments and what topographic, subsurface, and hydroclimatic conditions are related to these variations? By beginning to differentiate among the factors that control the transit of water through tropical mountain systems, which are rarely studied, we offer insights into how projected changes in precipitation patterns may interact with

the topography and subsurface conditions to shape water flow.

**2. Data and methods**

**2.1 Study area and sampling design**

In this study, we carried out detailed monitoring of seven small (areas ranging from 0.097−1.27 km²) catchments spanning the transition from the eastern flank of the Andes Mountains to the Amazon foreland floodplain (Figs. 1, 2; Table 1). The small catchments (SC) in this study are referred to by their sampling

point elevation in meters, followed by "-SC". Two small catchments (3472-SC and 3077-SC) are in the high Andes mountains, underlain by fractured shale bedrock, with mean slopes ~25−35 °. Two mid-

elevation small catchments (2432-SC and 1540-SC) are in the similarly steep mid-elevation Andes, with one (1540-SC) underlain by a granitic intrusion. One small catchment is situated in the foreland fold and thrust belt at the foothills of the Andes (609-SC), underlain by uplifted Andean sediments, with a mean

slope of 22.9 °. Two of the small catchments are situated on fluvial terraces in the foreland floodplain (276-SC and 214-SC), with the bedrock at these sites comprised of weathered sediments from the Andes. These catchments have much lower slopes, averaging 3–7 °. We also consider stable isotope data from two nested mesoscale catchments studied in Clark et al., 2014 (dashed white line in Fig.1b–d). Clark et al., 2014 used stream, precipitation, and cloud water isotope data to constrain a regional water budget for the mesoscale

Andean catchments. We reanalyzed their data to calculate stream young water fractions ($F_{yw}$) to complement the large amount of new data we report from our small study catchments across the elevation gradient. The catchments from Clark et al., 2014 are referred to by their mean elevation in meters, followed by "-Clark": 3195-Clark (mean slope 28.7 °; total area 49.8 km$^2$) and 2805-Clark (mean slope 31.6 °; total area 164 km$^2$). Site 3195-Clark drains Andean shales and site 2805-Clark drains Andean shales and the

same granitic intrusion that underlies 1540-SC (Fig. 1d).

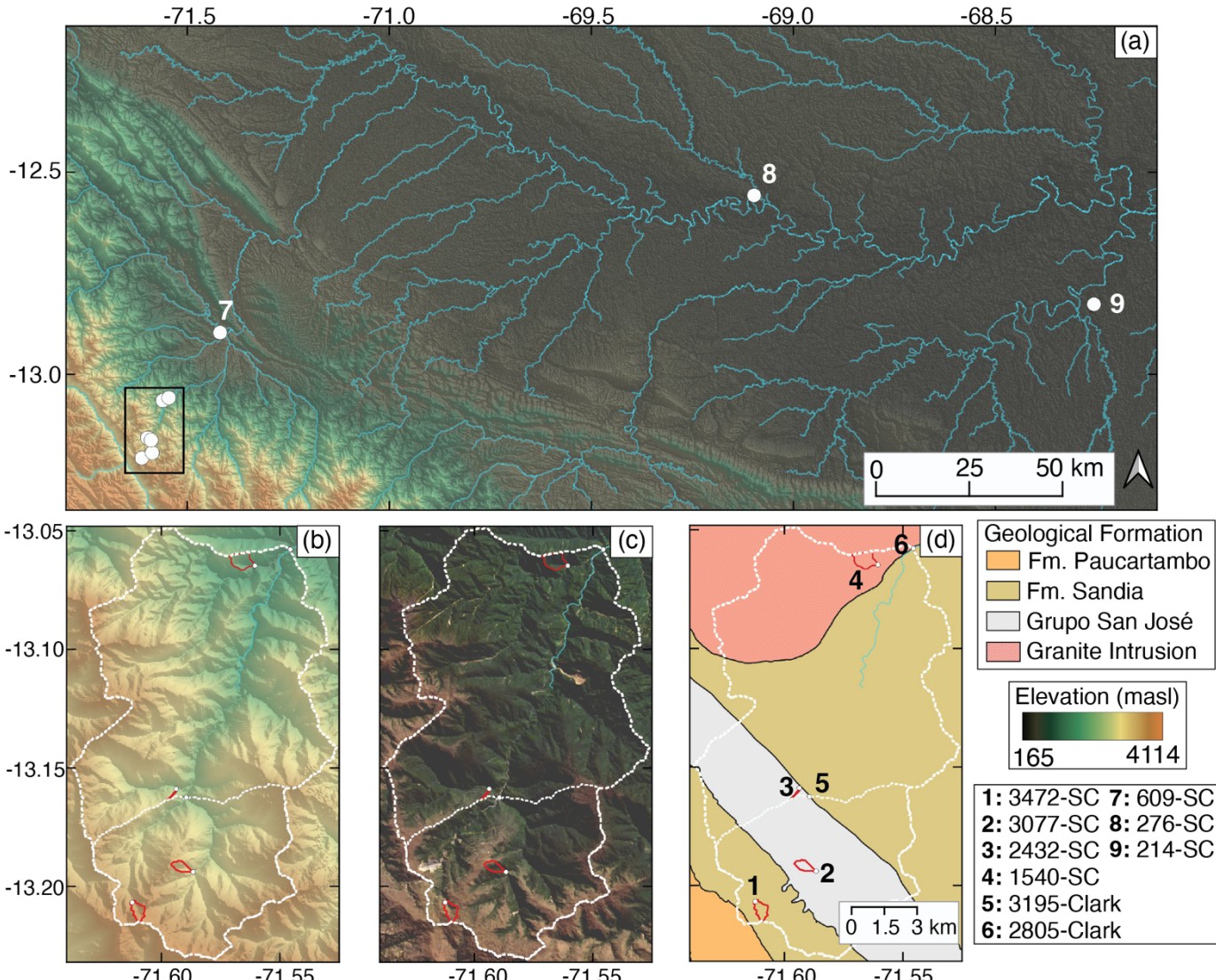

**Figure 1. (a) Digital Elevation Model (DEM, from ALOS 30m data) of the Andes mountains and Amazon floodplain in southern Peru. The stream network in (a-d) is for display only and does not represent the full extent of perennial streams in the study area. White circles indicate sampling locations. (b−d) show the area within the black rectangle in (a), with small catchments from this study delineated by solid red lines, and catchments from Clark et al., 2014 by dashed white lines. (b) shows elevation of Andean sites, (c) Landsat imagery, and (d) geology, using data from INGEMMET.**



**Table 1. Characteristics of small catchments from this study and mesoscale catchments from Clark et al., 2014. TMCF = tropical montane cloud forest, TMF = tropical montane forest, TRF = tropical rainforest.**

| Site | Latitude | Longitude | Area (km²) | Mean slope (°) | Geology | Soil* | Vegetation |
|---|---|---|---|---|---|---|---|
| 3472-SC | -13.2062 | -71.6117 | 0.552 | 25.6 | Shale (Sandia Fm.) | na | Puna |
| 3077-SC | -13.1926 | -71.5880 | 0.326 | 34.6 | Shale (San José group) | Umbrisol | TMCF |
| 2432-SC | -13.1597 | -71.5938 | 0.097 | 26.4 | Shale (San José group) | na | TMCF |
| 1540-SC | -13.0645 | -71.5604 | 0.203 | 37.7 | Granite Intrusion | Cambisol | TMF |
| 609-SC | -12.8961 | -71.4183 | 0.163 | 22.9 | Interbedded sandstone & shale (Cabanillas group) | na | TRF, Bamboo** |
| 276-SC | -12.5588 | -70.0993 | 0.412 | 6.9 | Fluvial terrace (Quaternary) | Ultisol | TRF |
| 214-SC | -12.8296 | -69.2713 | 1.270 | 3.8 | Fluvial terrace (Quaternary) | Cambisol | TRF |
| 3195-Clark | -13.1628 | -71.5892 | 49.800 | 28.7 | Shale (Sandia Fm. & San José group), | na | Puna, TMCF, TMF |
| 2805-Clark | -13.0603 | -71.5444 | 164.300 | 31.6 | Shale (Sandia Fm. & San José group), Granite Intrusion | na | Puna, TMCF, TMF |

**\*(Asner et al., 2017; Wu et al., 2019; Pitman et al., 2001), \*\*Secondary growth rainforest dominated by bamboo (Wu et al., 2019). na = data not available.**

The seven small streams were sampled approximately bi-weekly beginning in April 2016. In addition to stream sampling, precipitation was collected at sites 3077-SC, 1540-SC, 609-SC, 276-SC and 214-SC. For sites 3472-SC and 2432-SC we calculated approximate precipitation oxygen isotope values by linearly interpolating between nearby precipitation samples collected at higher and lower elevations, supported by the observation that in this region precipitation isotopes have a linear relationship with elevation (Ponton et al., 2014). Precipitation was collected in a bucket left out between each sampling, with a layer of oil to prevent evaporative loss. Point discharge was manually measured each time a sample was taken. For sites 3077-SC and 609-SC, continuous discharge was measured in 2019 and 2020 with WL16 Global Water

Level Loggers. Rainfall amount data are from tipping bucket and Vaisala rain gauges maintained by the

Andes Biodiversity and Ecosystem Research Group, a manual rain gauge maintained by the Los Amigos

Biological Station, and rain gauges operated by the Servicio Nacional de Meteorología e Hidrología del

Perú (SENAMHI). Stream baseflow indices were calculated for sites 3077-SC and 609-SC using the

Matlab HydRun hydrograph analysis package (Tang and Carey, 2017).

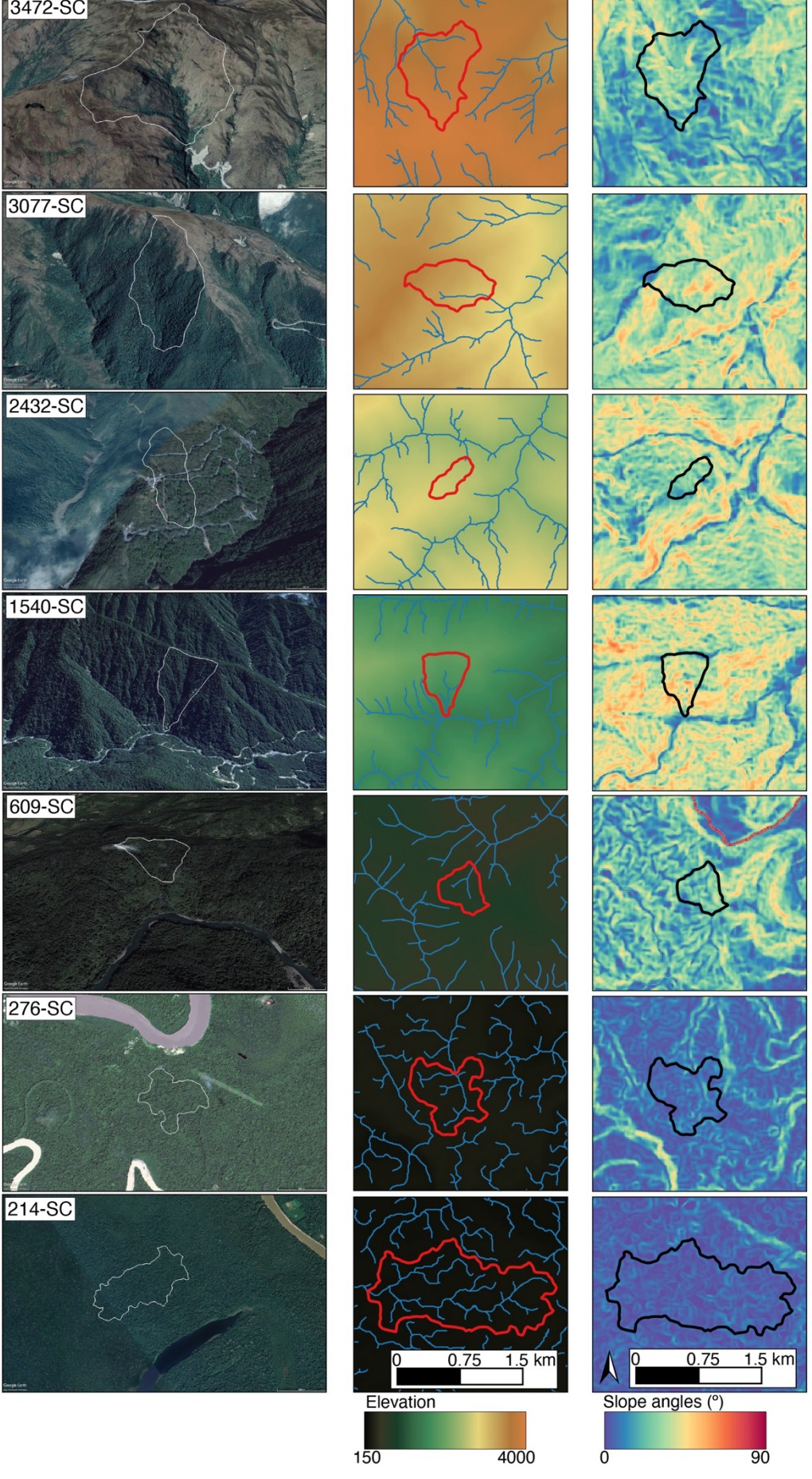

**Figure 2. Satellite imagery with catchment boundaries in white (first column; imagery: Google Earth, Image © 2023, CNES/Airbus and Maxar Technologies), elevation maps with catchment boundaries in red and smoothed stream network in blue (second column), and maps showing hillslope angles with catchment boundaries in black (third column) for each of the seven small catchments in this study. Elevation and slope angles derived from smoothed 8m DEM generated from stereo pairs of WorldView satellite images using the SETSM algorithm (see text).**

## 2.2 Topographic analyses

We determined catchment topographic parameters based on digital elevation models (DEMs) generated from stereo-pairs of WorldView satellite imagery using the SETSM algorithm (Noh and Howat, 2015). We produced smoothed DEMs covering the studied catchment areas at 8m horizontal spatial resolution using a combination of cross-track and in-track image pairs (several different image pairs were needed to cover all the study catchments). Topographic calculations were completed in GrassGIS (v8.2) using available algorithms (GRASS Development Team, 2022). Catchment areas were determined using multiple flow direction (MFD) routing for all except the Tambopata site where only single (D8) flow routing yielded a physically reasonable hydrologic representation. Stream networks (as shown in middle panel of Figure 2) were defined based on a threshold flow accumulation area of 12,500 $m^2$, which captured observed streams in the smallest catchment (2432-SC). A single threshold area is probably not appropriate across such diverse terrain but was adopted in our calculations for consistency. For each catchment, three metrics of flow path length were calculated: the distance to the catchment outlet along flowpaths (using r.stream.distance; Jasiewicz and Metz, 2011), the distance along flowpaths to the stream network defined by the 12,500 $m^2$ accumulation threshold (also using r.stream.distance), and the length of all flow path vectors defined by one raster cell spacing between each flow line, calculated using r.flow. Because of the very low slope angles on the terrace surfaces, flowline computation was not possible for the two lowland catchments, 276-SC and 214-SC (SETSM-derived DEMs generally capture topographic metrics well, but noise may be relatively more pronounced in flatter terrain; e.g., see Atwood and West, 2022). Flowpath gradients were calculated using r.stream.slope, and catchment-wide slope angles with r.slope.aspect. Importantly, key topographic metrics such as slope angles and hillslope lengths are dependent on elevation model resolution as well as threshold values used to define stream channels (e.g., see comparison in Clark et al. 2016 for the region in this study). Consequently, care should be taken in making any comparison with analogous values from other studies.

## 2.3 Stable isotope data analysis


Samples were analyzed for stable isotopes of water ($\delta^{18}$O and $\delta$D), with results reported here using permille notation relative to the Vienna Standard Mean Ocean Water standard. The stream oxygen or hydrogen isotope composition is referred to as $\delta^{18}$O$_{stream}$ and $\delta$D$_{stream}$ and precipitation oxygen and hydrogen isotope composition as $\delta^{18}$O$_{precip}$ and $\delta$D$_{precip}$. The analyses were carried out via two Los Gatos Research Liquid

Water Isotope Analyzers (LGR) (Caltech and Lawrence Berkeley National Lab) and a Picarro L2130i Cavity Ring Down Spectrometer (Chapman University). The internal error of isotope measurements on the Picarro was 0.1 ‰ or better for $\delta^{18}$O and 2 ‰ or better for $\delta$D. On the LGR at Lawrence Berkeley National Lab the internal error was 0.1 ‰ or better for $\delta^{18}$O and 1 ‰ or better for $\delta$D. On the LGR at Caltech the internal error was 0.3 ‰ or better for $\delta^{18}$O and 1 ‰ or better for $\delta$D. Long-term accuracy on certified

isotope standards was within one standard deviation of the known isotopic values.

Young water fractions were calculated for each small catchment following Kirchner (2016a, 2016b). Stream and precipitation oxygen isotope data were fit with Equation (1):

$$C(t) = a_s \times \cos(2\pi f t) + b_s \times \sin(2\pi f t) + k \qquad (1)$$

where C is the concentration of a tracer in stream or precipitation, t is time, f is the frequency of the interval, a and b are the cosine and sine coefficients and k is the vertical shift. The fit to stream and precipitation isotope data was performed with and without stream discharge and rainfall amount weighting. The young water fraction was then calculated using Equations (2-4):

$$A_{stream} = \sqrt{a_s^2 + b_s^2} \qquad (2)$$

$$A_{precip} = \sqrt{a_p^2 + b_p^2} \qquad (3)$$

$$F_{yw}\,(\%) = A_{stream}/A_{precip} \qquad (4)$$

where A is the amplitude of the seasonal cycle in stream and precipitation oxygen isotopes. In addition to calculating young water fractions using all of the stream water isotope data, we divided isotope data from each site into quartiles according to observed stream runoff and calculated young water fractions (again

using Equations 1–4) for each quartile of stream runoff (*sensu* Gallart et al., 2020b).

In order to assess the uncertainty of the young water fraction estimates, we bootstrapped observations of stream and precipitation isotope values for each site. For each bootstrap resampling, we drew one sample at random from the complete dataset for each catchment and then repeated this resampling from the complete

dataset until we had drawn the same number of random samples as the original dataset (e.g., for a dataset with 50 observations, we sampled 50 times, each time from the full dataset). We repeated this process

10,000 times for each stream and precipitation isotope dataset at each site. We then calculated $F_{yw}$ from equations 1-4 for each of those 10,000 bootstrapped datasets. This procedure allowed us to constrain the young water fraction as a distribution of values for each site.

For comparative purposes, we also generated a null dataset of young water fractions across all sites. To do this, we first took individual stream and precipitation isotope values from each site and subtracted each observation from the site's mean stream and precipitation isotope value. This approach allowed us to normalize for differences in stream and precipitation isotope values across sites due to orographic effects. We combined the site-specific normalized stream and precipitation isotope datasets to create null datasets containing normalized stream (n = 394) and precipitation (n = 257) isotope values for all sites. We then bootstrapped with replacement from the complete dataset, as described above, by drawing 394 observations from the stream dataset and 257 observations from the precipitation dataset. We repeated this 10,000 times each for the null stream and precipitation datasets and calculated $F_{yw}$ from equations 1-4 for each of those 10,000 bootstrapped datasets.

## 3. Results

### 3.1 Rainfall and stream discharge

The frequency of storms and the amount of precipitation vary across the Andes Mountains to Amazon floodplain transition due to orographic effects. Sites 214-SC and 276-SC have mean annual precipitation of ~3200 mm (Rapp and Silman, 2012) and 2479 ± 275 mm (Amazon Conservation Association, unpublished data), respectively. As moisture moves across the foreland floodplain and reaches the Andean foothills and mid-elevation mountains, mean annual precipitation increases: at 872 m (closest to site 609-SC) mean annual precipitation is 5371 ± 507 mm (SENAMHI). The highest precipitation is measured in the mid-elevation mountains: at 1379 m mean annual precipitation is 10425 ± 1214 mm. At 2161 m the mean annual precipitation is 8325 ± 687 mm. Precipitation amounts then decrease towards the high-elevation mountainous sites: At 2912 m mean annual precipitation is 4110 ± 418 mm. Site 1540-SC has the largest and most frequent rains (Fig. 3a), consistent with it being one of the sites with the greatest mean annual precipitation. Site 609-SC has the second most frequent and most intense rains, while the high-elevation 3077-SC and foreland floodplain 276-SC have less frequent and less intense rains. The changes in hydroclimate from mountain foothills to high-elevation mountains are also reflected in stream discharge patterns (Fig. 3b); specifically, Site 609-SC has a flashier hydrograph and lower baseflow index than site 3077-SC.

The discrete nature of the stream sampling and limited time resolution of our sample collection could introduce bias in estimation of $F_{yw}$ (Gallart et al., 2020b). As one check on how representative our sampling was of flow conditions, we compared mean stream runoff corresponding to times of sample collection with

mean stream runoff from the continuous runoff records for sites 609-SC and 3077-SC. For site 609-SC the mean discharge during sample collection was 8.8 mm/d while the mean discharge of the continuous record was 8.0 mm/d. For site 3077-SC the mean discharge during sample collection was 11.6 mm/d while the mean discharge of the continuous record was 11.0 mm/d. The similarity in the mean values may reflect the low discharge variability at our tropical study sites compared to catchments in temperate and

Mediterranean climates, yet even in this setting, incomplete sampling across the flashy hydrograph is expected to introduce uncertainty in calculated $F_{yw}$ values.

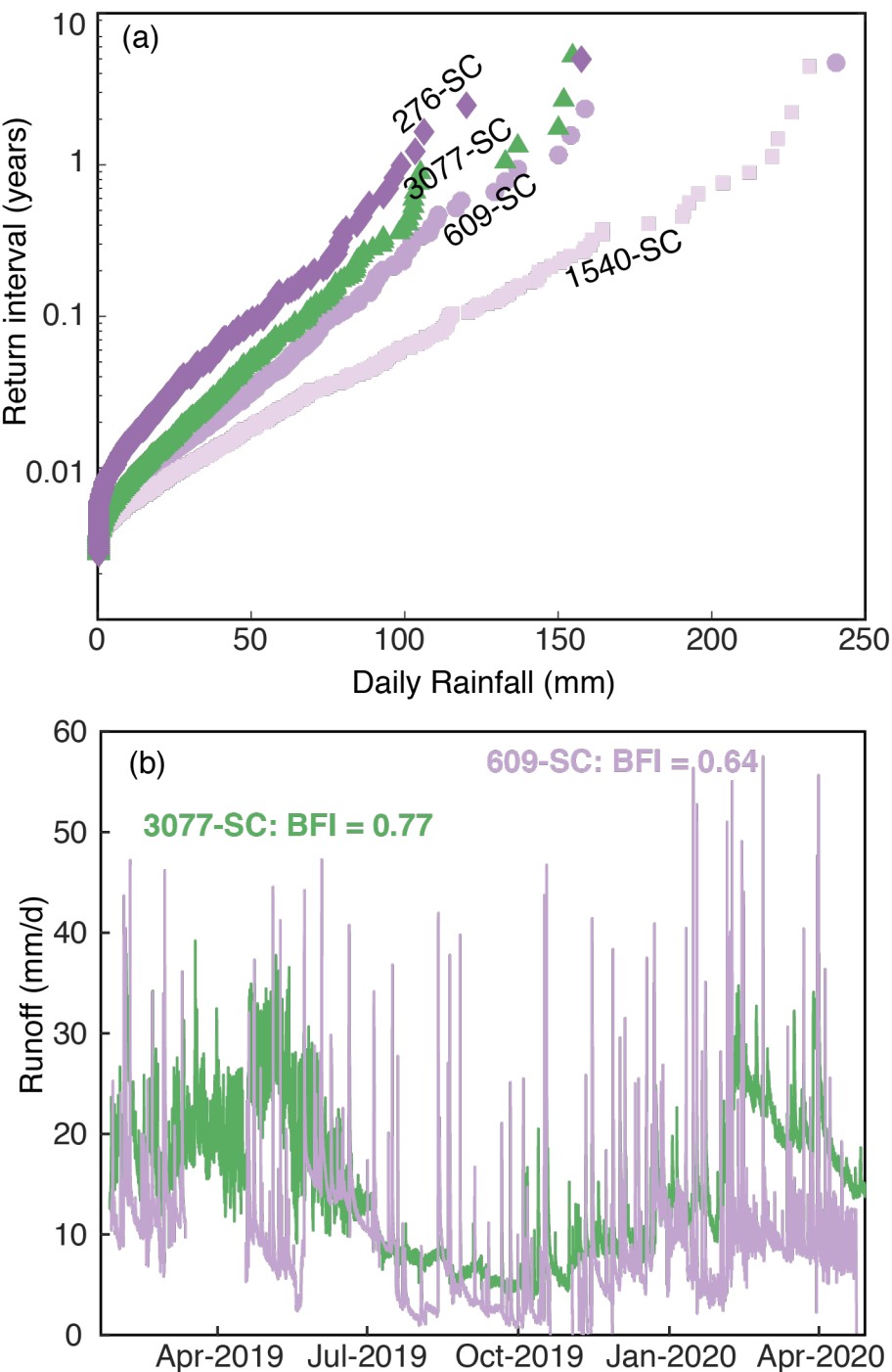

**Figure 3. (a) Precipitation return interval for rain gauges near sites 3077-SC (ABERG Wayqecha Rain Gauge), 1540-SC (ABERG Trocha Union Rain Gauge), 609-SC (SENHAMI Chontachaca Rain Gauge) and 276-SC (ACCA Los Amigos Rain Gauge). (b) 3077-SC and 609-SC stream runoff records with baseflow indices (only these two sites had reliable continuous discharge records).**

**3.2 Oxygen and hydrogen isotopes in streamflow and precipitation**

The $\delta^{18}O_{stream}$ and $\delta^{18}O_{precip}$ values follow an orographic trend across the transition from high Andes
mountains to foothills (3472-SC to 609-SC), with the highest elevation streams showing the most isotopic
depletion (Figs. 4 –6). The $\delta^{18}O_{precip}$ trends with elevation are similar to those reported previously for this
region (e.g., Ponton et al., 2014), here adding seasonal information. Our new data reveal that along this
same mountain-to-foothill transition, $\delta^{18}O_{precip}$ and $\delta D_{precip}$ display marked seasonal cycles (amplitude
$\delta^{18}O_{precip}$ ~ 4–5 ‰). The average volume-weighted $\delta^{18}O_{precip}$ seasonal cycle amplitude is greatest in the high
Andes mountains (4.3 ‰) and mid-elevation mountains (4.5 ‰) and lowest in the mountain foothills (3.7
‰) and foreland floodplain (3.2 ‰) (Table 2; Figs. 4, 6).

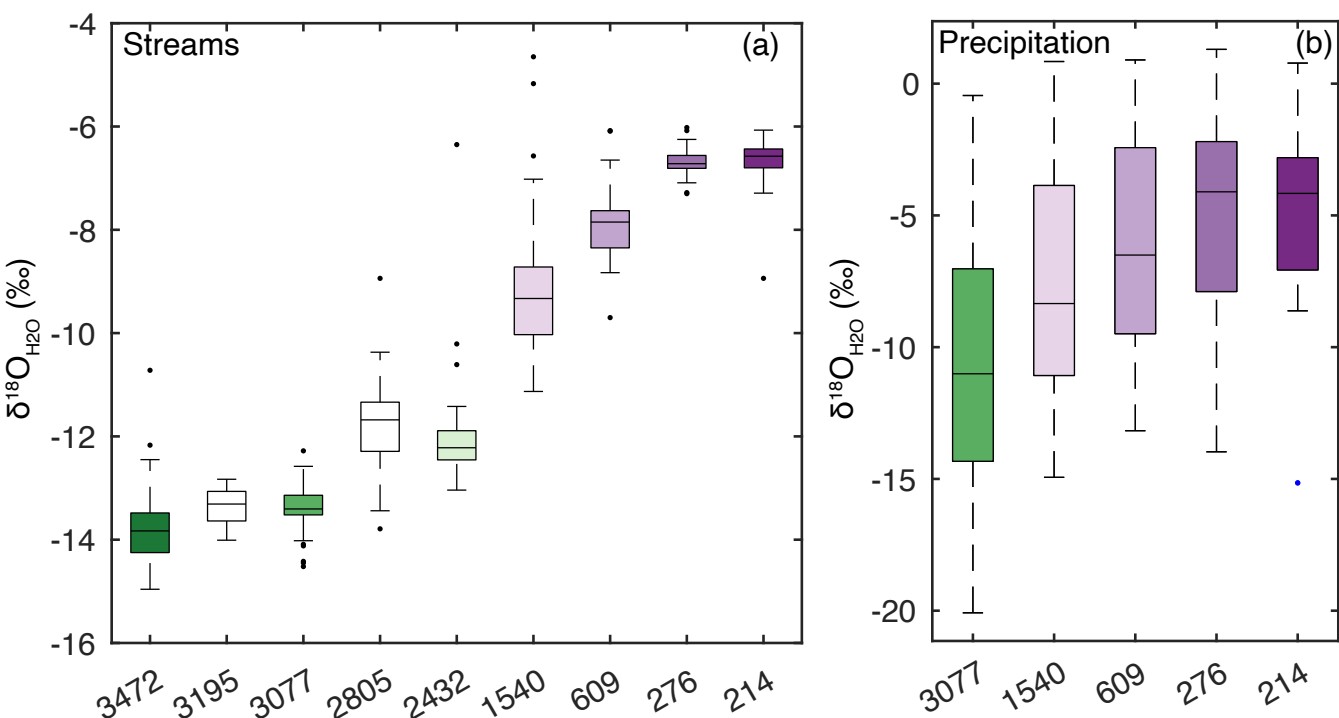

**Figure 4. (a) $\delta^{18}O_{stream}$ for small catchments and mesoscale catchments 3195-Clark and 2805-Clark**
**(from Clark et al., 2014; boxes left uncolored). (b) $\delta^{18}O_{precip}$ for precipitation collected near five of the**
**small catchments. Colors range from dark green to dark purple as a function of elevation. Boxes**
**show the interquartile range, whiskers show the non-outlier maximum and minimum and circles**
**indicate outliers.**


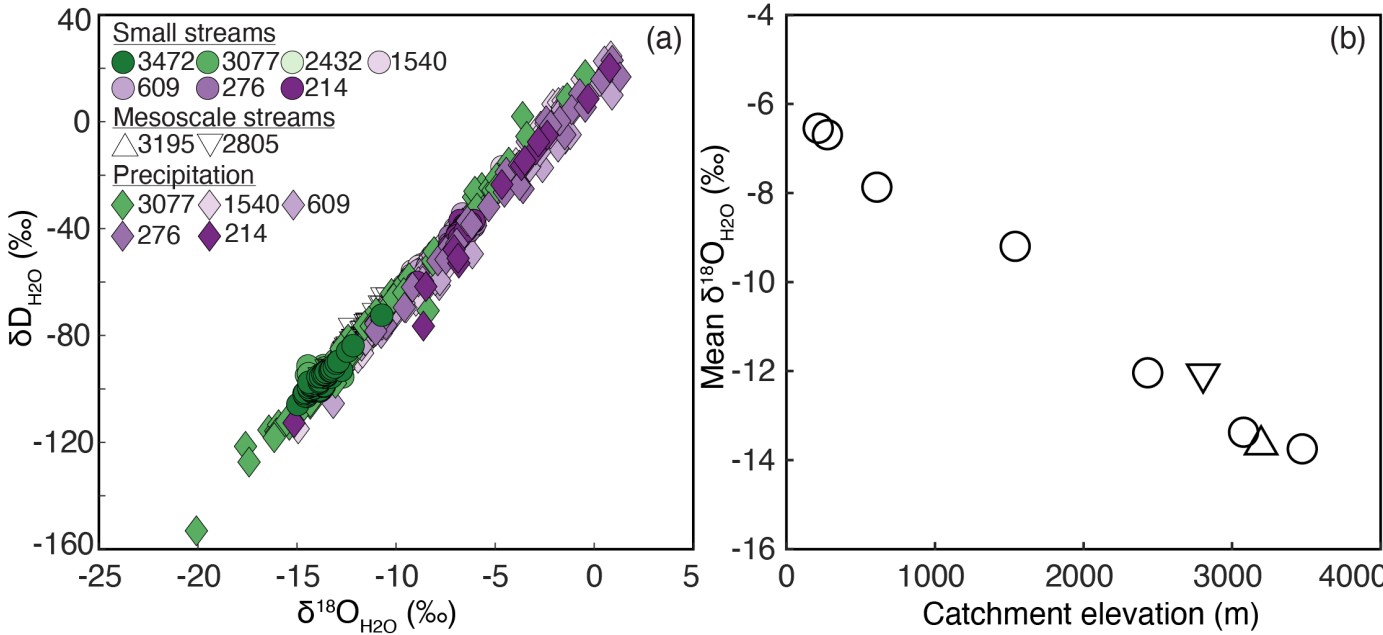

**Figure 5. (a) $\delta^{18}O$ and $\delta D$ of stream and precipitation. Colors for the small catchment data points range from dark green to dark purple as a function of elevation. (b) mean $\delta^{18}O_{stream}$ (not flow-weighted) as a function of sampling point elevation for small catchments and mean catchment elevation for mesoscale catchments. Circles represent small catchment stream isotopes, triangles show mesoscale catchment stream isotopes (from Clark et al., 2014) and diamonds show precipitation isotopes.**

Relative to the $\delta^{18}O_{precip}$ inputs, $\delta^{18}O_{stream}$ values are damped. The degree of isotope dampening and therefore the amplitude of the $\delta^{18}O_{stream}$ seasonal cycle varies between the small catchments situated from mountain-to-foothill (Fig. 5). The seasonal amplitude of $\delta^{18}O_{stream}$ values is smallest within the Andes mountains (3472-SC, 3077-SC, 2432-SC) and foreland floodplain sites (276-SC and 214-SC) and highest for the mid-elevation mountain (1540-SC) and mountain foothills sites (609-SC) (Fig. 5). Of the two mesoscale catchments, 3195-Clark has a smaller seasonal amplitude in $\delta^{18}O_{stream}$ than 2805-Clark. Dual isotope space ($\delta^{18}O$ and $\delta D$) reveals no significant deviation from the local meteoric water line (Fig. 5), indicating no significant evaporative signal in the stream waters for any of the sites in this study.

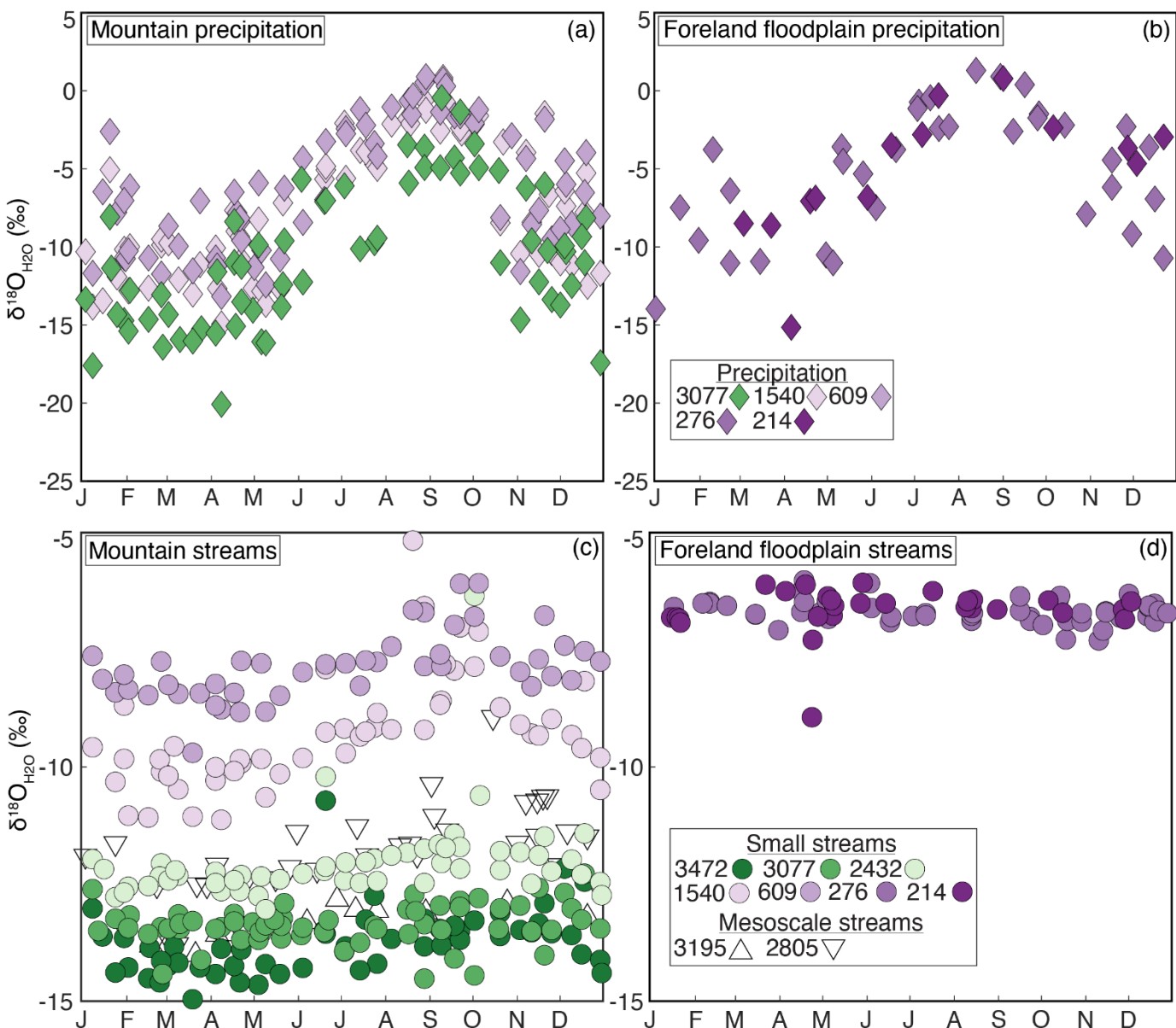


**Figure 6.** δ¹⁸O$_{stream}$ (a and b) and δ¹⁸O$_{precip}$ (c and d) for the duration of the study period (2016−2020), plotted by day of year. Refer to Fig. 6 for the fit of Equation 1 to data. Circles represent δ¹⁸O$_{stream}$ from the small catchments; triangles represent δ¹⁸O$_{stream}$ from the mesoscale catchments. Diamonds represent δ¹⁸O$_{precip}$. Panels (a) and (c) show sites in the Andes and mountain foothills; panels (b) and

(d) show the foreland floodplain sites.


**Table 2. Discharge and rainfall-weighted oxygen isotope seasonal cycle amplitude with propagated standard errors ($A_{stream}$ and $A_{precip}$), derived from Equations 1-3. $R^2$ values are included for the fit of Equation 1 to stream and precipitation isotope data. N = number of stream and precipitation stable water isotope data from this study and Clark et al., 2014. Dashes (-) indicate locations without precipitation collection. For sites without precipitation collection, $\delta^{18}O_{precip}$ was linearly interpolated by elevation from the nearest sites. *Due to the limited data for site 214-SC, only unweighted seasonal cycle amplitudes and $F_{yw}$ were calculated.**

| Small catchments | Location | $n_{stream}$ samples | $A_{stream}$ (‰) ± SE | $R^2$, stream | $n_{precip}$ samples | $A_{precip}$ (‰) ± SE | $R^2$, precip |
|---|---|---|---|---|---|---|---|
| 3472-SC | Mountain | 56 | 0.60 ± 15 | 0.22 | - | 4.60 ± 0.54 | 0.54 |
| 3077-SC | Mountain | 52 | 0.19 ± 0.18 | -0.02 | 74 | 4.19 ± 0.50 | 0.49 |
| 2432-SC | Mountain | 55 | 1.10 ± 0.24 | 0.28 | - | 4.25 ± 0.49 | 0.55 |
| 1540-SC | Mid-elevation mountain | 53 | 2.32 ± 0.21 | 0.71 | 68 | 4.46 ± 0.47 | 0.57 |
| 609-SC | Mountain foothills | 46 | 0.74 ± 0.11 | 0.55 | 66 | 3.75 ± 0.53 | 0.43 |
| 276-SC | Foreland floodplain | 54 | 0.22 ± 0.05 | 0.21 | 34 | 3.19 ± 0.96 | 0.21 |
| 214-SC | Foreland floodplain | 28 | 0.17 ± 0.15* | -0.03 | 15 | 5.36 ± 1.02 | 0.69 |
| *n samples total* | | 344 | | | 257 | | |
| Mesoscale catchments | Location | $n_{stream}$ samples | $A_{stream}$ (‰) ± SE | $R^2$, stream | $n_{precip}$ samples | $A_{precip}$ (‰) ± SE | $R^2$, precip |
| 3195-Clark | Mountain | 17 | 0.45 ± 0.09 | 0.62 | - | 4.48 ± 0.57 | 0.55 |
| 2805-Clark | Mountain/mid-elevation mountain | 33 | 1.17 ± 0.19 | 0.54 | - | 4.34 ± 0.55 | 0.55 |
| *n samples total* | | 50 | | | | | |


### 3.3. Young water fractions

Young water fractions ($F_{yw}$) vary between the catchments across the mountain-to-floodplain transition. Fig. 8 shows the distribution of unweighted and flow-weighted $F_{yw}$ obtained from the bootstrap resampling routine. Our bootstrap resampling offers one way of addressing the uncertainty associated with young water fraction calculations and demonstrates the reliance of $F_{yw}$ on the inclusion (or exclusion) of a sampling event. 3472-SC, 3077-SC and 2432-SC have flow-weighted $F_{yw}$ between 5 and 25 %. Mesoscale catchment

3195-Clark, draining approximately 50 km$^2$ of Andean shales, has a flow-weighted $F_{yw}$ of 10 %, on the lower end of the range of flow-weighted $F_{yw}$ seen in the three small Andean catchments. At mid-elevations, 1540-SC, which drains granitic intrusions, has a flow-weighted $F_{yw}$ of 52 %. The second mesoscale catchment, 2805-Clark, which drains a 165 km$^2$ area including Andean shales and granitic intrusions, has a flow-weighted $F_{yw}$ of 27 %. 609-SC, in the foothills of the Andes and underlain by colluvium, has a flow-weighted $F_{yw}$ of 22 %. 276-SC, located on a foreland floodplain fluvial terrace, has a flow-weighted $F_{yw}$ of 7 %. For comparison, the null dataset, generated from a compilation of isotope data from all sites, yields $F_{yw}$ of 7 %. In addition to changes in the mean values across the Andes-Amazon gradient, the distributions from the bootstrap resampling routine change across the region, with wider distributions for the mid-elevation catchments and tighter distributions in the high Andes and Amazon lowland catchments.

Figure 9 shows the relationship between key catchment characteristics and the weighted best estimate of stream young water fraction (Fig. 9). There is no apparent relationship between $F_{yw}$ and slope angle or flow path length, calculated either as median distance to the stream network or the average length of all flow paths (Fig. 9b,c,d). There is, however, some coherent pattern in $F_{yw}$ across these catchments, including a "humped" relationship with elevation (Fig. 9a) and a positive relationship with mean annual precipitation from the nearest rain gauges (Fig. 9e). Additionally, all the sites (except for 276-SC, most likely due to a small number of sampling points) point to an increase in the stream young water fraction with increasing discharge quartiles (Fig. 7). These observations may help to explain the decoupling of $F_{yw}$ and topography at these sites, as explored below.

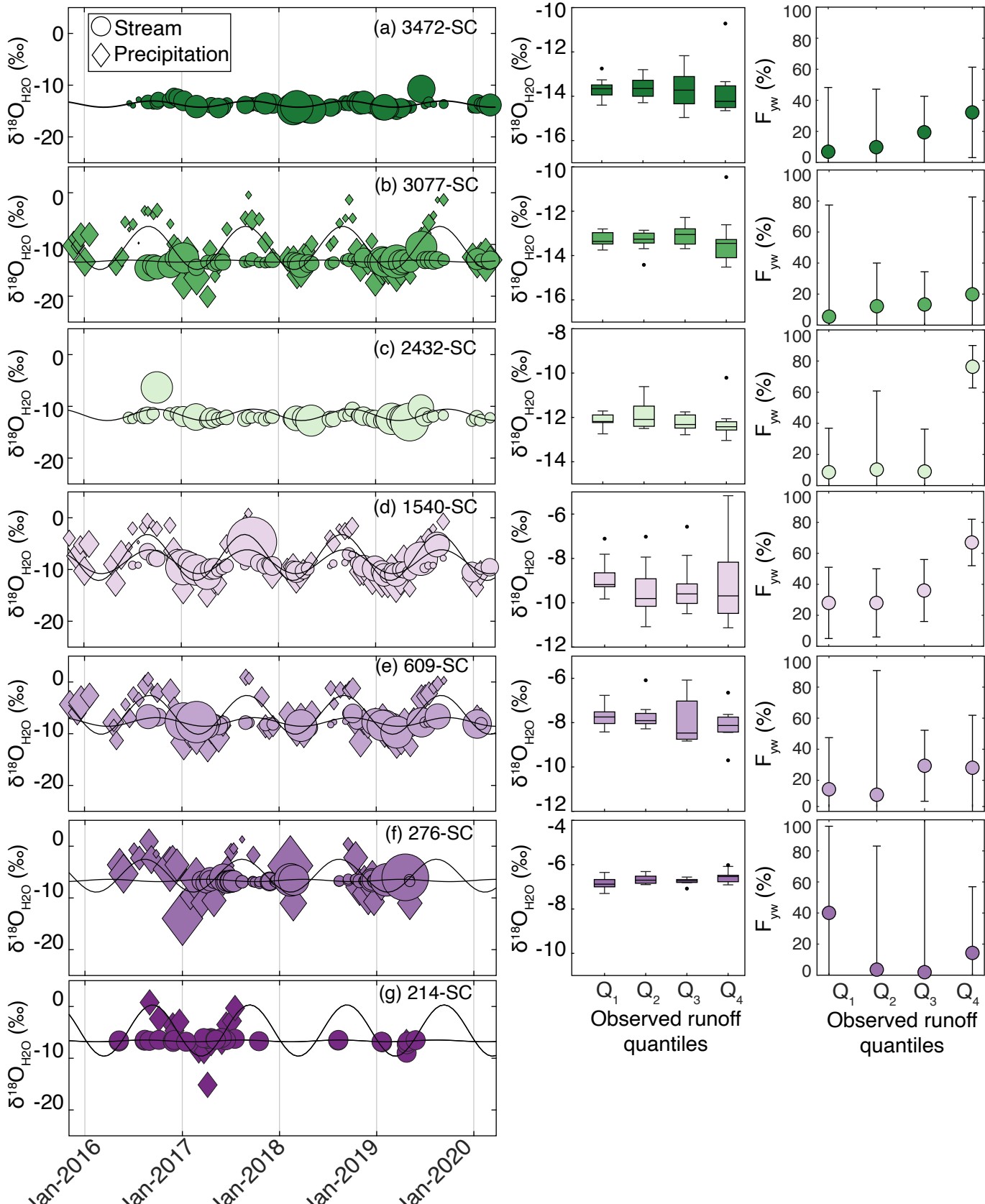

**Figure 7.** Left panel: $\delta^{18}O_{stream}$ (solid circles) and $\delta^{18}O_{precip}$ (open diamonds) from twice monthly sampling campaigns in each small catchment. The circles and diamonds are scaled using runoff and rainfall weights. Solid black lines indicate the weighted fit of Eqn. 1 to $\delta^{18}O_{stream}$ and $\delta^{18}O_{precip.}$ Middle panel: $\delta^{18}O_{stream}$ distributions across observed quartiles ($Q_1 = 0$–25 %, $Q_2 = 25$–50 %, $Q_3 = 50$–75 %, $Q_4 = 75$–100 %) of runoff for each site. Right panel: young water fractions calculated for each quantile of observed runoff values; note large uncertainties preclude extensive interpretation except to note that differences between catchments persist across all ranges of discharge. Data in (a−c) are from small catchments in the mountains, (d) is from the mid-elevation mountain small catchment, (e) is from the foothill small catchment and (f and g) are from the foreland floodplain small catchment. Limited data from 214-SC prevented calculation as a function of runoff quartiles.

405

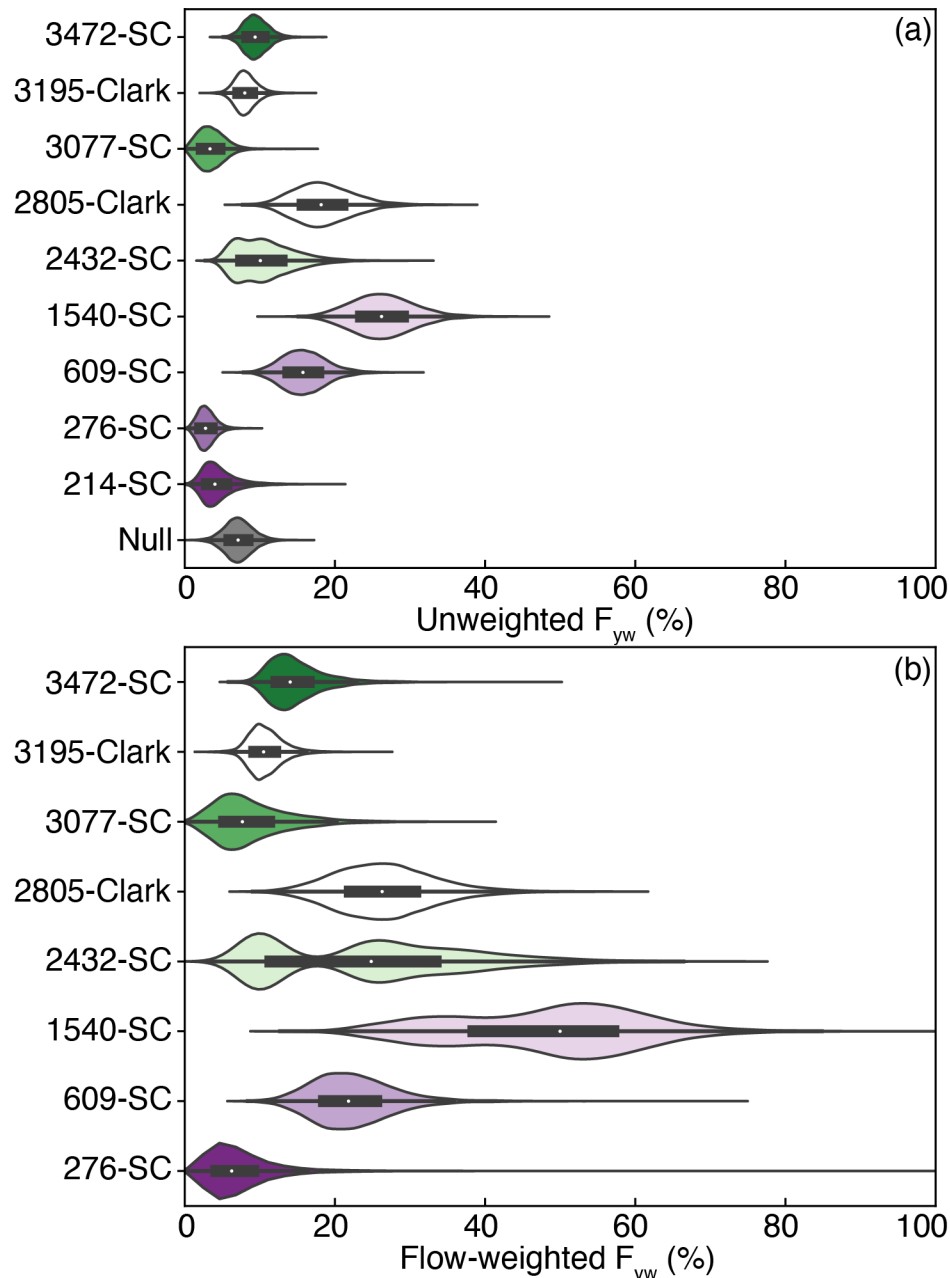

**Figure 8. Unweighted (a) and weighted (b) stream young water fractions for all catchments and a null dataset.**

410

415

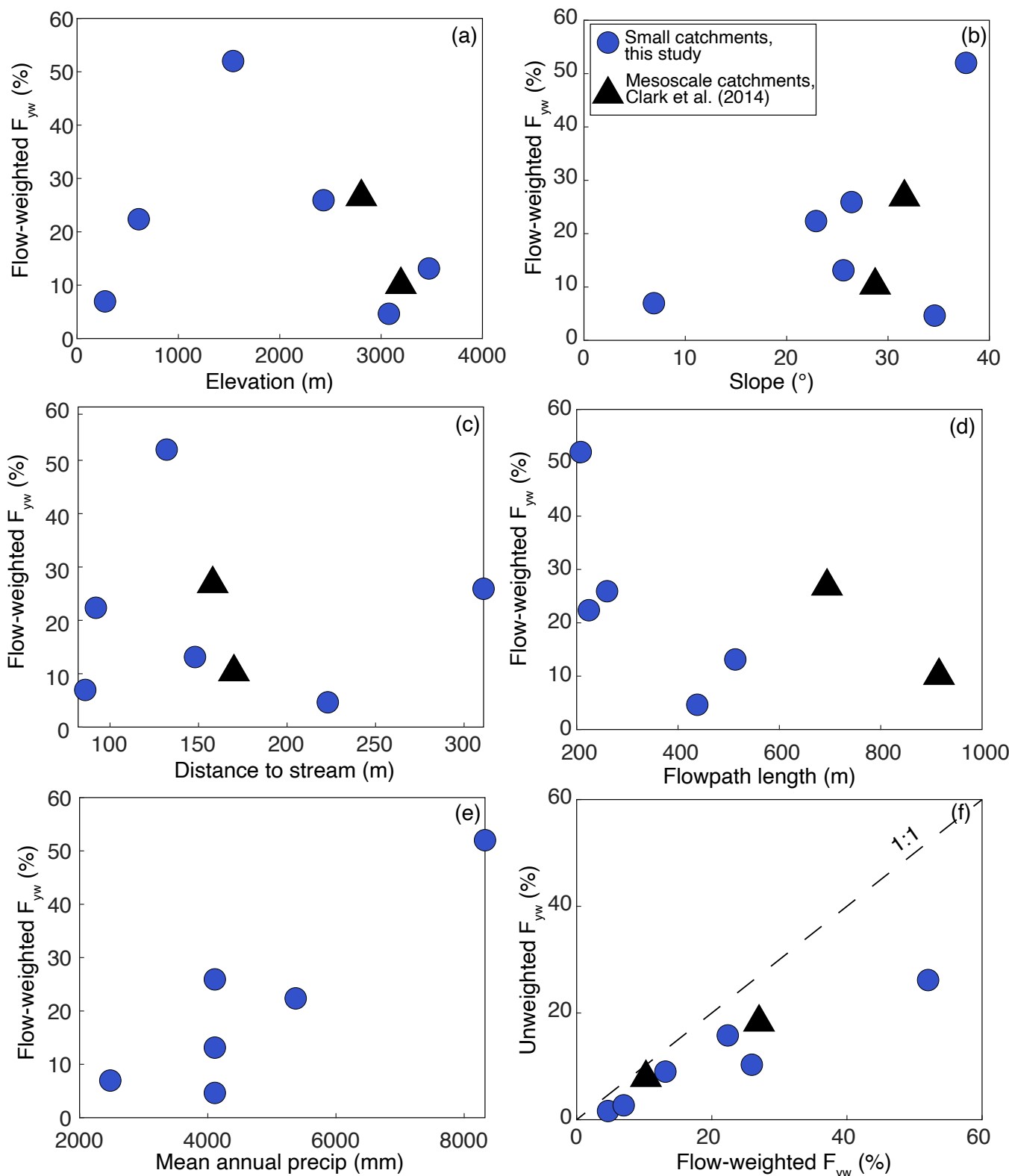

**Figure 9. Circles represent small catchments from this study, triangles represent mesoscale catchments from Clark et al. (2014). Flow-weighted $F_{yw}$ as a function of (a) catchment elevation at sampling point for the small catchments, and mean catchment elevation for the mesoscale catchments, (b) mean catchment slope, (c) median distance to the stream network along flow paths,**

**(d) median length of all flow paths from their origin to the catchment outlet, and (e) mean annual precipitation at the nearest rain gauge to each catchment. (f) Compares flow-weighted mean $F_{yw}$ to unweighted mean $F_{yw}$.**

## 4. DISCUSSION

### 4.1 Hydroclimate and permeability control stream young water fractions

The small catchments in the high Andes Mountains (3472-SC, 3077-SC and 2432-SC) all have low $F_{yw}$, with flow-weighted $F_{yw}$ between 5 –26 % and relatively tight distributions resulting from the bootstrap resampling routine. The mid-elevation small catchments show a wider range of bootstrap $F_{yw}$ values, tending toward much higher $F_{yw}$ (Fig. 8). A larger range of $F_{yw}$ values obtained from the bootstrap resampling routine indicates that the $F_{yw}$ is sensitive to the inclusion or exclusion of points that lead to higher $F_{yw}$ (i.e., streams with greater amplitude in $\delta^{18}O$ have more variable isotope values that lead to a wider spread of $F_{yw}$ values). The $F_{yw}$ values inferred from the mesoscale catchments studied by Clark et al. (2014) are consistent with the patterns from the small catchments. The mesoscale catchment in the high Andes, has a similar $F_{yw}$ to that of the high elevation small catchments (flow-weighted $F_{yw}$ = 10 %). In contrast, the mesoscale catchment that spans across the high- to mid-elevations (2805-Clark) has a flow-weighted $F_{yw}$ of 27 %, consistent with a mixture of water with low $F_{yw}$ from upstream portions of the study region and water with high $F_{yw}$ from the mid-elevations. Overall, our data point to low and tightly distributed $F_{yw}$ in the high mountains, but higher and more broadly distributed $F_{yw}$ in the mid-elevations.

The $F_{yw}$ values found this study are broadly similar to those from other mountainous regions previously studied but with some potentially interesting differences. Mountainous watersheds in Germany were found to have a flow-weighted median $F_{yw}$ of 13 % and a 10-90[th] percentile range of 4-17 % (Lutz et al., 2018), while Swiss Alpine streams had flow-weighted median $F_{yw}$ of 20 %, and a 10–90[th] percentile range of 10–38 % (von Freyburg et al., 2019). In comparison, the high Andean sites in this study have flow-weighted $F_{yw}$ of 5-26 %, at the lower end of these other studies, while the mid-elevation mountainous sites (1540-SC and 2805-Clark) had somewhat higher flow-weighted $F_{yw}$ (52 % and 27 %, respectively) than found previously. Potential reasons for these differences are discussed below. At larger spatial scales, Jasechko et al. (2019) found a global flow-weighted mean $F_{yw}$ of 34 % and 10-90[th] percentile range of 4-53 %. The Andes-Amazon results all fall within these global ranges, although the lowland sites have amongst the lowest $F_{yw}$ observed anywhere (flow weighted $F_{yw}$ = 7 %, unweighted $F_{yw}$ 3%).

We attribute the low $F_{yw}$ observed in the high mountain sites in our study at least in part to high permeability of the fractured shale bedrock. Fractures create conduits for fluid flow that can be magnified by dissolution of reactive minerals, such as the sulfides that are relatively abundant in the Paleozoic shale underlying our Andes Mountains catchments. Previous studies of stream hydrochemistry in the region have emphasized the importance of sulfide mineral oxidation as a primary weathering process (Burt et al., 2021; Torres et al., 2016), and pyrite oxidation is known to generate porosity and permeability in shale bedrock (Gu et al., 2020). In our conceptual model of water transit, the combination of pore-scale chemical weathering and regional stresses create a fractured subsurface that is conducive to long fluid flow paths, leading to overall low young water fractions in Andean streams.

The mid-elevation catchments differ in two respects that we think can explain the distinct transit times inferred for these streams. The higher $F_{yw}$ values for watersheds between 2805 and 609 m coincide with a shift to a flashier hydroclimate: there are more rainfall events of higher magnitude at the mid-elevations compared to either the high Andes or the Amazon lowlands (Fig. 3a; also see Clark et al., 2016). Correspondingly, the stream hydrograph at 609-SC is flashier than at 3077-SC (Fig. 3b; these are the two catchments with a semi-continuous discharge record). A comparison of stream baseflow indices for sites 3077-SC and 609-SC shows a higher baseflow index for site 3077-SC (BFI = 0.77) and lower baseflow index for site 609-SC (BFI = 0.64). We interpret the first-order shift in $F_{yw}$ values from the high Andes (where baseflow indices are high) to the mid-elevations (where baseflow indices are lower) as being related to this change towards a wetter, stormier climate, suggesting a primary role for hydroclimate forcing in determining transit times in these mountainous catchments. An important role for precipitation and discharge regimes has emerged from other recent transit time studies focused on single catchments with higher temporal resolution data collection (Gallart et al., 2020b; von Freyberg et al., 2018b; Stockinger et al., 2016). We see some slight variability in the amplitude of $\delta^{18}O_{stream}$ as a function of discharge in our results (Fig. 6, following Gallart et al., 2020b). These trends are consistent with our interpretation that precipitation regime plays a key role in determining water transit times at our sites, but we lack data across the range of discharge that would be needed for robust quantitative analysis of this effect. Higher frequency sampling across gradients such as those in the Andes, though daunting given the logistical challenges of this environment, would be an interesting target for future work.

Superimposed on the overall differences that characterize the mid-elevation catchments, the $F_{yw}$ in 1540-SC stands out as especially high (Fig. 9; flow-weighted $F_{yw}$ >50%). Unlike the other catchments in our study that are characterized by sedimentary bedrock, this catchment is underlain by a granitic intrusion (Clark et al., 2014). The especially high $F_{yw}$ in this part of the study region may be related to the low permeability of

this granite bedrock, which would prevent water from infiltrating deeply and leads to rapid, near-surface flow paths over the steep topography. In their study in Oregon, USA, Hale et al. (2016) found that

catchments with low permeability bedrock showed much shorter water transit times than catchments with high permeability bedrock, as well as a stronger relationship between topography and water transit times. The high $F_{yw}$ values we find at site 1540-SC is consistent with the argument that low permeability bedrock and steep slopes contribute quickly deliver water to streams. Thus site 1540-SC highlights overlapping impacts of hydroclimate and lithology on $F_{yw}$ in this setting: this catchment has the highest $F_{yw}$, and a

combination of high total precipitation and low permeability granite bedrock. Yet with our present data, it is not possible to distinguish which variable (hydroclimate or lithology) exerts a stronger control on $F_{yw}$.

**4.2 Implications for the role of mountains in modulating water, erosional, and biogeochemical fluxes.**

The role of mountains as water towers, and particularly the response of these freshwater resources to climate change, depends in part on water transit times through mountain catchments. In revealing the importance of hydroclimate in determining transit times through mountain catchments, our results suggest that shifting precipitation regimes may be important in determining not just how much precipitation falls over mountain regions (or indeed the balance of snow and rain), but also the fate of precipitation as it

makes its way through mountain catchments. If our spatial comparison of catchments across the Andes-Amazon region translates to temporal trends, then a flashier rainfall regime in the future might be expected to produce a wider range of transit times including higher young water fractions in streams draining mountainous terrain.

Our observation of higher young water factions at catchments with more precipitation echo other studies showing that young water fraction increases at higher discharge (Gallart et al., 2020b; von Freyberg et al., 2018a). In this sense, our results are also consistent with recent studies suggesting that catchments can amplify rainfall variability (Müller Schmied et al., 2020) and that the young water fraction can vary on interannual timescales and is vulnerable to extreme weather events (Stockinger et al., 2019). The

implications for downstream flooding and the buffering of droughts may warrant further consideration.

The hydrology of mountainous catchments may play important geological roles, too. River discharge, and particularly discharge variability, exerts a primary control on erosion (e.g., Tucker and Bras, 2000). Longer transit times may dampen the relationship between precipitation variability and the river incision that drives

mountain erosion; systematic relationships between topography and water transit times could therefore either dampen or amplify erosional efficiency of a given precipitation regime. Catchment hydrology has

also been invoked as central to the role of mountain building in the global carbon cycle over geologic timescales (Maher and Chamberlain, 2014). This argument depends on both the exposure of fresh minerals for chemical weathering by rapid erosion, as well as systematic changes in hydrologic flow paths

associated with mountain building. The mountainous sites within this study display a wide range of values in flow-weighted $F_{yw}$ (from ~5–52 %; Fig. 9), with no systematic relationship between topography and $F_{yw}$. Although a global compilation of stream $F_{yw}$ shows a general negative correlation between topographic relief and $F_{yw}$ (Jasechko et al., 2016), that relationship is notably weak — and the $F_{yw}$ from the small catchments studied here emphasize how other environmental factors (hydroclimate, catchment architecture)

play important roles in determining the $F_{yw}$ of streamflow. Moreover, when comparing across the high Andes and Amazon lowlands, there is remarkably little difference in $F_{yw}$ despite dramatic differences in topography: catchments with average slope angles of ~5° and ~35° have similar flow-weighted $F_{yw}$: 5 % for site 3077-SC and 7 % for site 276-SC. This result argues against a systematic shift in water transit times associated with mountain building, but rather a variable response modulated by climatic and geologic

factors — although our results do point to a wider range in $F_{yw}$ associated with mountains than lowlands, at least for the tropical setting of the Andes-Amazon system.

While our results, and especially the $F_{yw}$ of lowland catchments, may be specific to the Andes-Amazon setting, we expect the hydroclimatic and geological effects that we document here to be more generally

relevant in other mountainous regions, too. Orographic controls on precipitation tend to force the highest precipitation, as well as the most intense rainfall, along mountain fronts and at mid-elevations. In addition to the Andes, similar patterns have been shown in the Himalaya (Bookhagen and Burbank, 2006) and the European Alps (Napoli et al., 2019), and models predict complex spatial patterns of orographic precipitation that depend on several factors including climatic variables (e.g., Barros and Lettenmaier,

1994; Roe and Baker, 2006). The dependence of catchment transit times on hydroclimate, as we find in the Andes and as reported in other recent work (von Freyberg et al., 2018b; Gallart et al., 2020b), suggests that orographic effects on rainfall regime may be a primary determinant of hydrologic processes in major mountain ranges. Similarly, we expect fractured bedrock, and associated high permeability, to be generally characteristic of mountain systems as seen in our work and other studies (e.g., Muñoz-Villers et al., 2016;

Moon et al., 2017), though our results also highlight how the geological complexity of mountains – such as the presence of a granitic intrusion in our study area of the Andes – can introduce heterogeneity. Full understanding of the role of mountainous regions in water, sediment, and geochemical cycles will depend on evaluating the role of these multiple factors in determining hydrological behavior.

## 5. Conclusions

We collected stream and precipitation samples for analysis of O and H stable isotope ratios at seven streams and four rainfall stations spanning the Andes-Amazon gradient. Samples were collected approximately twice monthly over a period of four years. The calculated stream young water fraction varied significantly between sites. For the highest elevation sites (3472-SC, 3077-SC and 2432-SC), flow-weighted young water fractions varied between 5–26 %. For the mid-elevation small catchments (1540-SC and 609-SC), flow-weighted young water fractions were higher, at 22–52 %. Catchments in the foreland floodplain had low young water fractions of 7 % when flow weighted and 2–3 % when not weighted.

We suggest that the low young water fractions for the Andean catchments are a result of long flow paths in fractured shale. High young water fractions observed at mid-elevation sites result from a stormier climate, and in the case of 1540-SC, granitic bedrock with poorly developed soils and low permeability, meaning that water moves through the catchment faster. In the lowlands, low permeability clay terraces and low relief together generate low young water fractions, highlighting the importance of the very low surface topography gradients in this setting. Thus a combination of topography, climate, and bedrock properties conspire to determine water transit across the Andes-Amazon transition. Our results emphasize the complexity of the role of mountainous regions in the hydrological cycle with more factors than just topography likely to control young water fractions at the global scale. Accounting for the multiple factors that control water transit will be important for fully understanding the role of mountain water towers in water, sediment, and carbon fluxes.

**Acknowledgements**

This work was funded by NSF award EAR-1455352 to AJW. We thank the Andes Biodiversity and Ecosystem Research Group (ABERG) for field support and access to rainfall data. ABERG rainfall data were collected with support of NSF DEB LTREB 1754647 to Miles Silman. We thank Alex Sessions and Fenfang Wu at Caltech, Markus Bill at Lawrence Berkeley National Lab and Fernando Silva at Chapman University for support with the stable isotope measurements. We thank Greg Goldsmith for support with stable isotope measurements and for helpful discussions. We thank Julien Emile-Geay for helpful discussions with respect to data analysis.

**Code/Data availability**

Stream and precipitation water isotope data as well as the Matlab code for young water fraction analysis and bootstrap resampling routine is available online via https://doi.org/10.4211/hs.c01ef51ca2b3495785d0f24c62142e23 (Burt et al., 2023).

## Author contribution

EB and AJW designed the study with input from DHCR and AJCQ. EB, DHCR and AJCQ carried out the monitoring. EB analyzed the samples and did the data analysis with input from AJW. AJW and AA carried out the topographic analyses. EB and AJW wrote the manuscript.

## Competing interests

The authors declare no competing interests.

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
