# Peer review of "Isotope-derived young water fractions in streamflow across the tropical"

_Hydrology and Earth System Sciences, 2022_

## Referee Comment (RC2)

[referee-annotated manuscript omitted]

---

## Author Comment (AC1)

**Referee #1**

*The manuscript "Hydroclimate and bedrock permeability determine young water fractions in streamflow across the tropical Andes mountains and Amazon floodplain" by E.I. Burt et al. proposes an analysis of the young water fractions in a set of small and mesoscale catchments in an area without or with scarce previous information.*

*The subject is timely and the area is poorly known, the paper is well structured and written and the figures and tables are adequate.*

*Nevertheless, the methods are rather outdated because the authors follow the approaches used in the early works when relevant aspects such as the importance of the sampling rate and the dependence of young water fraction on stream discharge were not yet sufficiently described. Some of the more significant papers in this aspect are cited by the authors in the discussion but not taken into account in the methods.*

*This means that the results obtained in this work are largely suspect to be dependent on the flow regime of the streams and on the moments when the samples were taken respect to the flow regime. The relationships obtained between the young water fractions and the characteristics of the catchments, stated from the title, may therefore be spurious.*

*In my opinion, the manuscript could be accepted for publication in HESS if the following recommendations are followed.*

**We thank the reviewer for their feedback. We are pleased that they recognize the timeliness and relevance of our work. Of course, we would have loved to collect higher time resolution data, as used in some recent papers estimating catchment young water fractions and their discharge-dependence — yet such sample collection was precluded by the extremely remote location and challenging conditions at our sample sites, as well as our focus on studying multiple catchments to capture differences between them. These factors are described more below.**

**We have evaluated discharge-dependence within our dataset, and while not as rich in information as we might have hoped (or might be possible with higher frequency sampling), the results do substantiate our interpretations of systematic differences between the catchments in our study, as described more below. We hope this clarification helps to allay the reviewer's concern that the relationships we identify may be artifacts of the sampling regime.**

*1) Given the scarce sampling rate and the lack of comparison with the flow regimes, the conclusions of the work should be removed from the title, which should be less conclusive.*

**We have updated the manuscript title to:**
"Isotope-derived young water fractions in streamflow across the tropical Andes mountains and Amazon floodplain"

*2) The sampling scheme did not follow strict time intervals so the 'unweighted' young water fractions are not time-weighted but of uncertain significance. Therefore, I strongly recommend to use only the 'volume-weighted' (the usual term is flow-weighted) young water fractions in the text, discussion and conclusions while the unweighted water fractions can be shown in a table just for comparison.*

**We have changed "volume-weighted" to "flow-weighted" and now use only the flow-weighted Fyw unless explicitly stated otherwise. We still include the unweighted results in Figure 7a and 8d for reference.**

*3) The dependence of young water fraction on discharge (discharge sensitivity) should be analysed. This dependence has been scarcely investigated but may inform on the behaviour of the catchments and be more robust for catchment comparison, because it might be less dependent on the sampling scheme and more appropriate than the young water fraction in this work. This dependence may be stated for every catchment using the equation (6) in the Gallart et al (2020) paper already cited in the manuscript.*

**When we first read the Gallart et al (2020) paper, we were excited by the findings and curious to see if we could apply a similar method to our dataset. There were several factors that limited our ability to apply the referenced Equation 6 to our data:**

**1. Given the remote location of the studied watersheds, as well as the especially challenging field conditions (including high temperature and humidity that rapidly rusts metals and degrades electronics, diverse animals that chew tubing and cables, low sunlight at ground level), it was impossible to employ auto sampling devices. Collecting all samples by hand limited the ability to sample across many quantiles of flow for each stream, especially for these streams which were all > 1km (often more) from any permanent structures.**

**2. Additionally, given that we were interested in answering questions about how the geomorphic gradient of the Andes mountains to Amazon foreland floodplain influenced the stream young water fraction, we opted for a greater spatial resolution — sampling seven different watersheds — and ultimately sacrificing the high temporal resolution achieved by Gallart et al. (2020) for one watershed.**

**However, despite these limitations, we still attempted to apply the framework from the Gallart et al. paper early on in our work, to see if it would be useful. For 6 of the 7 studied watersheds, we had enough oxygen isotope data to divide into four quartiles, based on discharge. We then calculated the young water fraction for each of the quartiles of isotope data. The figure below shows the results. We had initially decided not to include these results in our manuscript due to the**

high uncertainties. And, with respect to the reviewer's comment, we still maintain that these uncertainties (and our limitation of calculating only across discharge quartiles) make application of Eq. 6 from Gallart et al. difficult.

Yet despite the high margins of error (due to the small number of samples in each quartile), there are some interesting points that emerge from this analysis, worth pointing out in the context of the reviewer's question. Given the interest in discharge-dependency of $F_{yw}$ as expressed by the reviewer, we have also added these plots to panels in Fig. 6, with ample caveats about the large uncertainties.

Key takeaways from this figure include the observation that all the sites (except for 276-SC, most likely due to a small number of sampling points) point to an increase in the stream young water fraction with increasing discharge quartiles — much as expected based on prior work (e.g., Gallart et al., 2020) — albeit with very large uncertainties that preclude unequivocal interpretation or further quantitative analysis of the discharge-dependency. Yet, importantly, the mid-elevation sites (especially Site 1540-SC) have the overall highest $F_{yw}$, and $F_{yw}$ increases consistently across the quartiles. $F_{yw}$ increases with discharge also for the high Andes sites (3472-SC and 3077-SC), but the values are systematically lower than for the mid-elevation 1540-SC site. Thus, the high overall $F_{yw}$ that we calculate at 1540-SC is not simply an artifact of discharge dependency but holds across all discharge quartiles. This consistency suggests that the differences we observe between catchments are not simply differences in discharge-dependency but rather reflect underlying hydrologic behavior of the catchment. Site 2432-SC has a very high $F_{yw}$ for the highest quartile, which is caused by several storms that had a distinct isotopic influence on the stream oxygen isotope composition. This effect explains the very large range in bootstrapped $F_{yw}$ values in Fig. 7b, and it is worth pointing out that the range in these bootstrapped values implicitly captures the effect shown in the quartile plots (e.g., similar values across the quartile plots will yield tight distributions in the bootstrapped results, and vice versa).

*A more detailed explanation of the resampling approach used is recommended*

We agreed with this comment and added a thorough explanation of the resampling approach at the end of section 2.2 (Lines 395-400; 440-45).

*Finally, the current knowledge shows that "it has been difficult to identify a simple topographic control on young water fractions at the global scale" because of the inadequate and variable sampling schemes and the lack of consideration of flow regimes. Different sampling schemes and periods can give different results (Stockinger et al. 2016 and 2019, Gallart et al., 2020). The authors should propose a final conclusion more adequate to the limited sampling schemes used in their work.*

The reviewer raises a fair point. It is not our purpose in this manuscript to assess all the factors that may affect results from prior studies. Yet we do think our

results highlight the potential complexity of catchment transit times in mountainous terrain and have rewritten this sentence to capture that point, as follows:

"Our results emphasize the complexity of the role of mountainous regions in the hydrological cycle, with more factors than topography likely to control young water fractions at the global scale."

*Suggested paper not cited in the manuscript: Stockinger et al. (2019). Time variability and uncertainty in the fraction of young water in a small headwater catchment. Hydrol. Earth Syst. Sci., 23, 4333–4347*

We thank the reviewer for drawing our attention to this paper. We have now cited this paper on line 1153.

---

## Author Comment (AC2)

*The paper uses a unique isotope dataset for seven tropical catchments, spanning a range in elevation and particularly slope characteristics to determine the effects of topography and geology on the young water fraction. Although the analyses are relatively straightforward (and perhaps somewhat limited), the study contributes to the still limited literature on the effects of topography, geology, and catchment wetness on transit times or young water fractions and the results are certainly of interest to the readers of HESS. Furthermore, it provides important information on a poorly studied region. The manuscript is well written and nicely illustrated (although the color scheme is not immediately clear and not explained).*

**We appreciate the reviewer's positive comments about what our study contributes and the value of the data we provide. In revision we have worked to clarify the color scheme used in the figures and hope this makes the paper more readable.**

*However, I also have several concerns. One major comment is that the most interesting parts of the results are given in the discussion section. These should be part of the results section. The other main comment is that the methods for the derivation of the topographic indices are not explained (what data source and algorithm were used?). In particular, I wonder if in this terrain a 30 m DEM is sufficient to calculate hillslope lengths or average flow path length to the stream. Finally, the introduction and the discussion sections could be strengthened by comparing the results more to the existing literature. I therefore recommend major revisions.*

*In addition to the main comments stated above and the more detailed list given below, I have also added some minor comments to the annotated pdf. Note that the editorial suggestions included in the annotated pdf are just suggestions.*

**We thank the reviewer for their suggestions. We have addressed each of these concerns as described below, in particular restructuring the Discussion (to move key text into the Results section as recommended), clarifying and revising the topographic calculations, and revising the Introduction.**

*The introduction is relatively general and could be stronger. One way to do this is to provide a bit more information on what the papers mentioned on L90-93 say about the effects of topography and geology (or soil type) on transit times or young water fractions.*

**We restructured the introduction with these comments in mind. The citations that were originally in lines 90-93 provided the basis for an expanded paragraph (now lines 88-104). We elaborated upon the original citations and added more about previous studies investigating the effects of topography, geology, and hydroclimate on stream young water fractions.**

*Provide more details on the Clark (2014) study, so that it is clearer how this study differs from that study.*

**We added the following clarification to lines 288-291:**

**"Clark et al., 2014 used stream, precipitation and cloud water isotope data to constrain a regional water budget for the mesoscale Andean catchments. Given advances in stable water isotope applications since the 2014 study, we reanalyzed their data to calculate stream young water fractions ($F_{yw}$) while also adding a large amount of new data from small catchments across the elevation gradient."**

*L134: Do you mean total area instead of mean area?*

**This was a typo; we thank the reviewer for catching it and have corrected the mistake.**

*Also, provide information on what algorithm and what data set were used to determine the topographic characteristics. I assume that you used a 30 m DEM. How dissected is the landscape and what are typical hillslope lengths? Can a 30 m DEM represent the hillslopes well or is it too coarse to calculate the distance to the nearest streams, i.e., are the hilsllopes and small streams smoothed out too much in this DEM? How was the location of the streams determined? What accumulated area threshold did you use for this? This affects all the distance to stream calculations – and probably also the hillslope length calculations. The stream network dataset shown in Figure 1 c-d seems to be insufficient for this job.I would also recommend to calculate a few other characteristics (e.g., those used by McGuire et al., Lutz et al., or McHale et al.,) and to show the relation (or lack thereof) with the young water fractions.*

**We appreciate the reviewer raising these points about our topographic indices and have taken several steps to improve these dimensions of our paper.**

**First, not including basic information about our topographic analysis in our Methods section was an oversight and is now corrected with the text provided below. This additional text answers the reviewer's multiple questions.**

**Second, we agree that our the 30m resolution of the DEM we used in our prior submission was borderline for calculating the metrics of interest. To address this shortcoming, we produced new DEMs from stereo-pairs of WorldView satellite imagery, using the SETSM algorithm to generate DEMs at 8m resolution. We have now used these elevation models for calculating the topographic metrics reported in the revised manuscript. In the end, the higher resolution does not change our results (e.g., the overall comparison of topography and Fyw values), but the new approach does offer more confidence that we have captured the distinction of hillslopes and streams.**

**Third, as suggested by the reviewer, we have now added more topographic metrics, similar to those reported in prior studies (see Figure 9). Of note, the flowpath length values in the original submission were determined as the total distance from the catchment outlet. This differs somewhat from the approach used in some prior work, where flowpath lengths were defined as the distance along flowlines before intersection with streams. In revision, this distinction has been clarified by using more specific terminology. Of course, neither metric is a perfect representation of the complex subsurface flowpaths. We also note that key topographic metrics such as slope angles and hillslope lengths are dependent on elevation model resolution as well as threshold values used to define stream channels (e.g., see comparison in Clark et al. 2016 for the region in this study). As a consequence, care should be taken in making any comparison with analogous values from other studies.**

**One final point of clarification — the stream network displayed in Figure 1 is for display purposes only, and in common with much map-making practice, does not correspond to the same network used in the morphometric calculations (displaying such a dense network at the scale of the maps in Figure 1 would be impractical). This point is now made in the manuscript including in the caption to Figure 1. We have also added a new figure (now Figure 2), which we hope better illustrates the contrast between the study catchments, and which does show the delineation of the stream network at the scale used in our quantitative analysis (with streams defined by a threshold flow accumulation area of 0.0125 km$^2$).**

**Text added regarding methods for topographic calculations:**

We determined catchment topographic parameters based on digital elevation models (DEMs) generated from stereo-pairs of WorldView satellite imagery using the SETSM algorithm (Noh and Howat, 2015). We produced smoothed DEMs covering the studied catchment areas at 8m horizontal spatial resolution using a combination of cross-track and in-track image pairs (several different image pairs were needed to cover all the study catchments). Topographic calculations were completed in GrassGIS (v8.2) using available algorithms (GRASS Development Team, 2022). Catchment areas were determined using multiple flow direction (MFD) routing for all except the Tambopata site where only single (D8) flow routing yielded a physically reasonable hydrologic representation. Stream networks (as shown in middle panel of Figure 2) were defined based on a threshold flow accumulation area of 12,500 m$^2$, which captured observed streams in the smallest catchment (2432-SC). A single threshold area is probably not appropriate across such diverse terrain but was adopted in our calculations for consistency. For each catchment, three metrics of flow path length were calculated: the distance to the catchment outlet along flowpaths (using r.stream.distance; Jasiewicz and Metz, 2011), the distance along flowpaths to the stream network defined by the 12,500 m$^2$ accumulation threshold (also using r.stream.distance), and the length of all flow path vectors defined by one raster cell spacing between each flow line, calculated using r.flow. Because of the very low slope angles on the terrace surfaces, flowline computation was not possible for the two lowland catchments, 276-SC and 214-SC

(SETSM-derived DEMs generally capture topographic metrics well, but noise may be relatively more pronounced in flatter terrain; e.g., see Atwood and West, 2022). Flowpath gradients were calculated using r.stream.slope, and catchment-wide slope angles with r.slope.aspect. Importantly, key topographic metrics such as slope angles and hillslope lengths are dependent on elevation model resolution as well as threshold values used to define stream channels (e.g., see comparison in Clark et al. 2016 for the region in this study). As a consequence, care should be taken in making any comparison with analogous values from other studies.

Jasiewicz, J. and Metz, M.: A new GRASS GIS toolkit for Hortonian analysis of drainage networks, Comput. Geosci., 37, 1162–1173, https://doi.org/10.1016/j.cageo.2011.03.003, 2011.

Noh, M.-J. and Howat, I. M.: Automated stereo-photogrammetric DEM generation at high latitudes: Surface Extraction with TIN-based Search-space Minimization (SETSM) validation and demonstration over glaciated regions, GIScience Remote Sens., 52, 198–217, https://doi.org/10.1080/15481603.2015.1008621, 2015.

*L144-147: What method did you use to get a catchment-average rainfall rate?*

**To clarify, we do not use (or report) catchment-averaged rainfall but rather use rainfall data from the closest precipitation monitoring stations to evaluate the rainfall regime including precipitation-weighted O-isotope seasonal cycles as well as the magnitude-frequency statistics. Additionally, we only observe significant orographic trends in rainfall for the Andes mountains sites. Given the lack of elevation gradient in the mountain foothills (site 609-SC) and the foreland floodplain (sites 276-SC and 214-SC), the intra-catchment rainfall variability applies only to the mountainous sites.**

**We emphasize that actual precipitation amounts do not influence our calculations of the flow weighted $F_{yw}$. In order to calculate the $F_{yw}$ weights, we divided the rainfall in each two-week interval during which precipitation isotope samples were collected by the total precipitation from the multi-year study. We do not believe it would be more accurate to calculate the catchment-wide average precipitation amounts for the sites where we see large ranges in elevation, given that we only use rainfall data to constrain the relative amounts of precipitation collected for each isotope sample.**

**To ensure that precipitation weights were similar across the elevation gradient, we plotted the weights for each of the Andean precipitation collection sites:**

[Figure]

The three precipitation collection sites show overlapping weights for the duration of the multi-year study. Given that there does not appear to be systematically different precipitation seasonality between the three Andes mountains precipitation collection sites (consistent with Rapp and Silman, 2012), we do not feel it would be appropriate (or correct) to apply any sort of elevation-based rainfall weighting correction.

We also recognize that elevation gradients within the catchments have likely caused isotopic gradients in precipitation. However, the young water fraction only relies on the *amplitude* of the seasonal cycle in precipitation oxygen isotopes, not the actual isotope values themselves. The amplitude of the seasonal cycle in precipitation oxygen isotopes is not significantly different for the two Andes mountains precipitation collection sites (4.19 ± 0.50 ‰ ($\delta^{18}$O) for 3077-SC and 4.46 ± 0.47 ‰ for 1540-SC). For this reason, we do not believe that the *amplitude* of the precipitation isotopes would vary within the watersheds.

*Section 2.1: Provide some information on the vegetation and soil type as well. I see that the vegetation in included in Table 1 but the difference between UPRF and TRF is not clear enough for a reader who is unfamiliar with this region.*

We have removed the phrase "upper rainforest" and replaced it with "tropical montane forest" to be more precise. We also include soil data as available from the literature.

*L191-194: I would recommend to rewrite the equations and to use symbols with super or subscripts in the equation, rather than words. Also is the double sin or cos in equation 1 is a typo.*

**The double sin & cos in Equation 1 is a typo. It has been fixed (and only appeared in the Microsoft Word document – the equations used in our code do not have any typos). Additionally, we rewrote Equations 2–4 with symbols instead of words, as suggested.**

*L197: For the resampling, what fraction of the data was excluded? Just one data point or more, like 20%?*

**We realize that the original description of the resampling was not clear, as also mentioned by Reviewer #1. We thoroughly edited the description of the resampling to make it easier to understand. We did not exclude any of the data, but instead used the following resampling approach, quoted from line 396:**

**"For each resampling, we drew one sample at random from the complete dataset and then repeated this resampling from the complete dataset until we had drawn the same number of random samples as the original dataset (e.g., for a dataset with 50 observations, we sampled 50 times, each time from the full dataset)"**

*L198-201: This part is not very clear. It would be good to rewrite it so that someone can repeat exactly what you did.*

**Agreed. We added a thorough explanation of the resampling approach at the end of section 2.2 (Lines 395-400; 440-45).**

*L213: How much is slightly greater and is this a statistically significant difference?*

**We appreciate the reviewer pointing out the imprecise language used in the previous version of our manuscript. We have removed the use of "slightly" greater and have taken several steps to understand the statistical significance of our data. First, we have added $R^2$ values to Table 2 to help assess the goodness of fit of Equation 1 to our oxygen isotope data. Additionally, we have included propagated standard errors for the stream and precipitation seasonal cycle amplitudes, also in Table 2.**

*Figure 2: The figures in the manuscript are all very nice but perhaps you can explain the color-scale in the caption or add a legend to this figure.*

**We thank the reviewer for this feedback. We have clarified that color scheme ranges from dark green to dark purple as a function of elevation in the caption of Fig. 3 (previously Fig. 2), where this scheme first appears.**

*L231-233: Describe the amplitudes and whether or not these are different for the*

*different streams. It would make sense to describe Figure 5 here and to give some information on the goodness of fits here. Are the fitted curves reasonable representations of the data? They are not great but look reasonable – but the goodness of fit is not quantified here. This makes it difficult to compare this with the fits in other studies on the young water fractions.*

**From above: we have added $R^2$ values to Table 2 to help assess the goodness of fit of Equation 1 to our oxygen isotope data. Additionally, we have included propagated standard errors for the stream and precipitation seasonal cycle amplitudes, also in Table 2.**

**We believe that the fits to the data are reasonable. However, we also note that the $R^2$ values for the fit of Equation 1 to stream oxygen isotopes are particularly dependent on the amplitude of the seasonal cycle. For example, site 3077-SC has one of the lowest seasonal cycle amplitudes ($\delta^{18}O$ = 0.19 ‰) and has a low $R^2$ value of -0.02. For data without a pronounced seasonal cycle, Equation 1 does not provide a more appropriate fit than a horizontal line. However, it does not mean that the fit is inappropriate, simply that the stream isotope data have a very low seasonal cycle amplitude.**

*Table 2: Are the amplitudes mentioned here the difference between the min and max values or the amplitudes of the fitted curves? Add some goodness of fit measure (see comment above).*

**We now clarify that the amplitudes we report are 1/2 the total annual amplitude. As noted above, we also report $R^2$ for each amplitude.**

*Section 3.2: Discuss figures 7 and 8 here.*

**We have removed Figures 7 and 8 from the Discussion and moved these into the Results section.**

*Discussion section 4.1: This section contains too many new results. Move these results into the results section (3.2). Move the rainfall and streamflow data (Figure 6) to either the study site description or create a new first results section for this. Then focus the discussion section more on the results. This includes more comparisons with other studies who have shown the relations between topographic characteristics and transit times or young water fractions (e.g., Lutz et al., McGuire et al., von Freyberg et al., etc.). Similarly add more discussion on papers that have looked at the effects of geology or soil types on mean transit times or young water fractions (e.g., Hale et al.; Soulsby and Tetzlaff, 2008).*

**We appreciate this feedback. We now introduce these data in the Results section. We have included the citations mentioned and added more discussion (Lines 950-965 and 1080-1100) to place our results into context with the relevant literature.**

*L307: How wide is this range compared to those found in other studies? Add some comparison.*

**See lines 950-965.**

*L307-308: This fits much better in the results section. It would strengthen the paper if this section was expanded.*

**We have moved lines 307-308 to the Results section and have focused the Discussion section more on placing the results into context with other literature, as suggested.**

*L312-314: How do these young water fractions compare to those in other studies or the global study by Jascheko? Add some comparison to the existing literature here.*

**See lines 950-965.**

*L336-341: The flashier hydrographs and differences in rainfall characteristics should already have been mentioned in the site description, but at the latest as the first section of the results section.*

**We created a new Results section 3.1 entitled "Rainfall and stream discharge" where we elaborate on the rainfall and discharge data.**

*Figure 6b: It is clear that these streams are very flashy. It is not clear over what range of the observed streamflow, you took the samples. Can you give some statistics for this, e.g., what part of the flow-duration curve cover your samples cover? Or split the figure so that it is possible to plot the d18O in this figure as well to see when samples were taken?*

**The following figure shows the exceedance probably of instantaneous runoff values (black circles) and the instantaneous runoff values at the time that samples were collected (yellow diamonds). Although we only have continuous runoff data for two sites (3077-SC in the high Andes mountains and 609-SC in the Andes mountains foothills), the available data are very useful. For site 3077-SC, it appears that we have almost completely captured the range of discharge with our sampling campaign, despite being unable to use autosampling devices (see discussion above on limitations to automating our work). The overall ranges in area-normalized runoff in the high Andes mountains 3077-SC (5~35 mm/d) are also notably lower than at site 609-SC (0~70 mm/d). This is consistent with our analysis of precipitation return intervals and stream baseflow indices (see updated Fig. 2 in the manuscript). Briefly, in Fig. 2 we show that precipitation frequency and amount are lowest in the high Andes mountains and Amazon foreland floodplain, and highest in the mid-elevation Andes mountains and mountain foothills. This is consistent with work from previous studies in the same region (e.g, Rapp and Silman, 2012).**

It is not surprising then, that we are able to capture nearly the full range of stream runoff values in the high Andes mountains site, while we miss some of the larger storms (30-70 mm/d) at site 609-SC. Of course, we would have been thrilled to use autosampling devices to capture the highest flows, but it was not realistic given the logistical challenges. However, we believe that if we were able collect more samples during high-discharge conditions, it would only make stronger the argument that we present for the hydroclimatic influences on the stream young water fraction — since more samples at the wettest sites 1540-SC and 609-SC would result in even high young water fractions. It appears that more samples for the high Andean sites are not particularly necessary to capture the full behavior of these systems, as we've managed to sample across most of the discharge and see a subdued relationship between discharge quantiles and stream young water fraction (see figure above).

[Figure]

*L386: Add some comparisons to the existing literature here. Suggested refs are provided in the annotated pdf.*

**We have added the references suggested by both reviewers to this section of the Discussion.**

*Figures 7 and 8 should be part of the results section, not the discussion section*

**Figures 7 and 8 are now added to the Results section.**

*L469: It is unclear from the discussion section (nor the study site description!) that the lowland sites have clay soils as well. Shouldn't this mean that the flow is fast as well? This should be discussed more clearly on L 357 where you discuss the fast flow and therefore higher young water fraction for the mid-elevation sites. Following the same*

*logic, shouldn't the low elevation sites then not have higher young water fractions as well? This could use some discussion.*

**We have added two paragraphs (Lines 1080-1130) to the Discussion talking about the role of watershed permeability and slope on the young water fraction (including relevant citations).**

---

## Author Response (AR2)

June 5th, 2023

Dear Dr. Hrachowitz,

Thank you for your time during the final stages of manuscript revision. We are pleased to report that we have taken care of the three minor revisions suggested by Referee #1. The final comments added clarity regarding the overlapping influence of hydroclimate and lithology on stream young water fractions. We also corrected a mislabeled axis. We feel that our manuscript is strong and are pleased to present a final version for publication.

Thank you again.

All the best,

Emily I. Burt
Postdoctoral Fellow
Chapman University
Orange, CA

;

1. *Stream discharge chronicles have usually log-normal distributions. In these distributions, the mode is smaller than de median and the median is smaller than the mean and these differences increase with the variance of the distribution. This means that any discrete samples will be closer to the mode than to the mean, and when averaged will give negatively biased discharge values. The larger the variance of the discharge and the smaller the number of samples, the more negative bias the sample average. If Fyw increases with discharge, this bias in discharge sampling propagates to Fyw sampling.*

*Along the bootstrap resampling method used to analyse the uncertainty of the Fyw estimates, an analysis of the likely role of infrequent sampling on Fyw biases should be applied to the two catchments with continuous discharge records shown in Figure 3b. The mean discharges of the 609-SC and 3077-S catchments should be calculated with all the available data and compared with those calculated with only discharges synchronous with sampling times. As these hydrographs suggest a large difference in discharge variance, this will provide with an idea of the representativeness of the samples taken respect to the real stream flow volumes, and the dependence of the likely bias on the discharge variance.*

*I do not expect the results of this exercise to contribute to any relevant modification of the results of the manuscript, but I do hope that the authors can discuss these results in a way to value the limitations of the sampling design, and to warn the readers about the likely errors that can be made when flashy streams are insufficiently sampled.*

**We agree that this is an interesting exercise to determine potential biases in sampling. We completed this exercise and found that the mean discharge during sampling was very similar to the mean discharge of the continuous hydrograph. One potential explanation for this is that the overall variability in the hydrograph in these tropical watersheds is smaller than that observed in some Mediterranean and temperate watersheds. We added the following text to the manuscript (lines 285–295) to highlight the importance (and difficulty) of sampling across the stream hydrograph:**

**The discrete nature of the stream sampling and limited time resolution of our sample collection could introduce bias in estimation of $F_{yw}$ (Gallart et al., 2020b). As one check on how representative our sampling was of flow conditions, we compared mean stream runoff corresponding to times of sample collection with mean stream runoff from the continuous runoff records for sites 609-SC and 3077-SC. For site 609-SC the mean discharge during sample collection was 8.8 mm/d while the mean discharge of the continuous record was 8.0 mm/d. For site 3077-SC the mean discharge during sample collection was 11.6 mm/d while the mean discharge of the continuous record was 11.0 mm/d. The similarity in the mean values may reflect the low discharge variability at our tropical study sites compared to catchments in temperate and Mediterranean climates, yet even in this setting, incomplete sampling across the flashy hydrograph is expected to introduce uncertainty in calculated $F_{yw}$ values.**

*2. The sentence "We interpret the first-order shift in Fyw values from the high Andes (where baseflow indices are high) to the mid-elevations (where baseflow indices are lower) as being related to this change towards a wetter, stormier climate, suggesting a primary role for hydroclimate forcing in determining transit times in these mountainous catchments" is in some contradiction with the preceding sentence "We attribute the low Fyw observed in the high mountain sites in our study at least in part to high permeability of the fractured shale bedrock".*

*In my opinion, the former is more consistent, given the wide range of precipitation, than those that attribute the observed differences in Fyw to lithological aspects. I suggest a rewrite of the text in order to indicate the difficulty of testing both hypotheses simultaneously and to avoid showing any preference on one of the two without adequate evidences.*

**We agree with the reviewer that determining if hydroclimate or lithology exerts a stronger control on young water fraction is very difficult. We have added the following text to line 510 of the manuscript:**

**Thus site 1540-SC highlights overlapping impacts of hydroclimate and lithology on $F_{yw}$ in this setting: this catchment has the highest $F_{yw}$, and a combination of high total precipitation and low permeability granite bedrock. Yet with our present data, it is not possible to distinguish which variable (hydroclimate or lithology) exerts a stronger control on $F_{yw}$.**

*3. The axes in Figure 9f should be interchanged.*

**We have corrected the axes in Figure 9F.**